# Long-Term Follow-Up of Renal Function in Children after Liver Transplantation—A Single Center Retrospective Study

**DOI:** 10.3390/children8080633

**Published:** 2021-07-25

**Authors:** Grzegorz Kowalewski, Piotr Kaliciński, Marek Stefanowicz, Ryszard Grenda, Piotr Czubkowski, Marek Szymczak

**Affiliations:** 1Department of Pediatric Surgery and Organ Transplantation, Children’s Memorial Health Institute, 04-730 Warsaw, Poland; p.kalicinski@ipczd.pl (P.K.); m.stefanowicz@ipczd.pl (M.S.); m.szymczak@ipczd.pl (M.S.); 2Department of Nephrology, Kidney Transplantation and Hypertension, Children’s Memorial Health Institute, 04-730 Warsaw, Poland; r.grenda@ipczd.pl; 3Department of Gastroenterology, Hepatology, Nutritional Disorders and Pediatrics, Children’s Memorial Health Institute, 04-730 Warsaw, Poland; p.czubkowski@ipczd.pl

**Keywords:** liver transplantation, chronic kidney disease, immunosuppression

## Abstract

Chronic kidney disease (CKD) is a common complication after liver transplantation (LT). Its prevalence with modern immunosuppression regimens, especially in children, is variable depending on the transplantation era. The study included 61 pediatric patients with at least 10 years of follow-up after liver transplantation remaining under constant care of the Department of Pediatric Surgery and Organ Transplantation. The analysis included several tests: estimated glomerular function (eGFR), results of screening for renal tubular defects and blood concentrations of basic immunosuppressive drug-tacrolimus. CKD was diagnosed in 3% of children at 12 years after LT. The maintaining of tacrolimus concentrations >4 ng/mL in long-term observation was associated with a significant increase of microalbuminuria. The presence of microalbuminuria, regarded as a risk factor of CKD, confirmed the necessity of regular comprehensive assessment of patients in long-term follow-up.

## 1. Introduction

The development of chronic kidney disease (CKD) in patients after liver transplantation (LT) is a problem known and described since the 1970s and 1980s [1]. Due to progress in transplantology and prolonged survival of patients, more attention is being paid to this problem, as it is particularly important in the pediatric population, in which 5-year graft/patient survival reaches 85–90% [2]. The prognosis of maintaining a kidney function after LT has been studied in children, and the variable outcomes have been shown. The prevalence of CKD up to 33% in long-term follow-up after LT was reported [3,4], and more optimistic data have also been published [5]. Early published data on the high incidence of CKD [6], related to chronic nephrotoxicity of calcineurin inhibitors (cyclosporine A and tacrolimus) currently should be reevaluated due to the advances in immunosuppression protocols, use of the modified drug formulas and dosing regimens, as well as the longer patient and liver graft survival. Our study has investigated the prevalence of chronic kidney disease in children who are long-time survivors after LT and analyzed the potential risk factors of kidney dysfunction in our population.

## 2. Patients and Methods

### 2.1. Patients

From 1990 to 2020, 828 pediatric liver transplantations (LT) were performed in Children’s Memorial Health Institute in Warsaw. The population of our study cohort was recruited from a living group of patients with more than 10 years of follow-up after liver transplantation, remaining under care of our outpatient clinic, willing to participate in our study and signing informed consent. Since, as a pediatric transplant center, we are only able to treat patients under 18 years of age, only transplantations carried out between 2001 and 2008 met the inclusion criteria. A planned evaluation was performed from 2017 to 2019.

### 2.2. Analysed Data

Research included a retrospective analysis of available medical data collected from liver transplantation throughout the whole observation period, including consecutive data recorded during follow-up. Initial analysis of available data before LT and at discharge included the following: recipient characteristics, indication for transplantation, pretransplant eGFR, type of liver failure (acute, chronic, acute-on-chronic), pediatric end-stage liver disease)/model for end-stage liver disease; PELD/MELD) scores, presence of renal failure defined as need for hemodialysis before and after LT, duration of anhepatic phase, primary immunosuppression regimen with mean values of consecutive through concentrations (C_0_) of calcineurin inhibitors (CNI). From the time of LT data from routine follow-up visits at 1, 5 and 10 years after LT were included in the study. At each visit, patient characteristics (height, weight, body mass index (BMI)), dose and drugs concentrations, laboratory data (eGFR, serum creatinine and urea concentrations, aspartate transaminase (AST), alanine transaminase (ALT)), international normalized ratio (INR) were analyzed. A planned analysis was performed during a visit, >10 years after LT, with a range of tests including patient characteristics (weight, height, BMI), laboratory data (eGFR, serum creatinine, urea, cystatin C, protein, albumin, urinary spot albumin concentration (UAC)), assessment of mean concentrations of immunosuppressive drugs.

### 2.3. Assessment of Renal Function

Glomerular filtration rate was estimated (eGFR) based on formulas using both serum creatinine (eGFRcr = k × height (cm)/serum creatinine (mg/dL); k = 0.413; according to the modified Schwartz formula; 2009) [7] and serum cystatin C (eGFRcys = 39.8 * (ht(m)/Scr) ^0.456^ (1.8/cysC) ^0.418^ (30/BUN) ^0.079^ 1.076 ^male^ (ht(m)/1.4) ^0.179^; Schwartz 2012) [8]. CKD (stage 3) was defined as a GFR below 60 mL/min per 1.73 m^2^ according to Kidney Disease Improving Global Outcome (KDIGO) [9] and based on a single estimation. Abnormal albuminuria in morning spot urine analysis was defined as >20 mg/mL.

### 2.4. Immunosuppression Protocol

Until 2006, patients received a triple immunosuppression regimen including tacrolimus (TAC), mycophenolate mofetil (MMF) and corticosteroids. Beyond this time point, a standard immunosuppression protocol included double regimen, including TAC and MMF, while corticosteroids have been limited to individual indications (ABO-incompatible LT, contraindications to MMF therapy, patients with autoimmune hepatitis (AIH) and/or primary sclerosing cholangitis (PSC)). Target TAC C_0_ concentrations were decreased from the range of 10–12 ng/mL directly after LT, to 8–10 ng/mL within 3 months, 6–8 ng/mL after 6 months and to 4–6 ng/mL >12 months after LT. MMF was discontinued after 6–12 months after LT in patients without evidence of an episode of acute rejection (AR).

### 2.5. Statistical Analysis

Statistica 12 software was used for statistical analysis. Data were presented as median and range or mean ± standard deviation (SD). Student unpaired t-test or Mann–Whitney U test were used for comparisons of continuous variables between two groups. The chi-square or the Fisher’s exact test was used for pairwise comparisons of categorical variables between the subgroups. The Pearson and Spearman correlations were used. The cumulative probability of remaining free from CKD was performed using Kaplan–Meier estimation. The primary end-point was eGFR < 60 mL/min/1.73 m^2^. Cox proportional hazard models were used to estimate the association between potential risk factors and lowering of eGFR < 90 mL/min/1.73 m^2^ in long-term follow-up. Since our goal was to evaluate long-term kidney function after LT, the creation of Cox regression model was based on patients with eGFR < 90 mL/min/1.73 m^2^ at >1 year after LT. Time to event was defined as a presence of eGFR values <90 at either 5, 10 or >10 years after LT. For the evaluation of prognostic factors, we used the Cox regression model for proportional hazards by computing hazard ratios (HR) and corresponding 95% confidence intervals (95% CI). For the multivariate model, we selected the variables that presented HR with a significance level ≤0.05, and those which, while not reaching the previous criteria, were considered relevant in the literature. P values less than 0.05 were considered statistically significant.

The study was conducted in accordance with the principles of the Declaration of Helsinki. The study was approved by the ethics board at the Children’s Memorial Health Institute (approval number: 24/KBE/2017). Because the data were reported using patient medical records, informed consent was signed in accordance with the above guidelines. The research was conducted within the framework of study no. S183/2019.

## 3. Results

### 3.1. Baseline Characteristics

The study included 61 pediatric recipients: 42 girls, 19 boys. The mean age at LT was 1.7 years SD ± 1.6, body mass 10.2 kg SD ± 4.4. Indications for transplantation were: biliary atresia in 40 patients, liver tumors in 4 patients, toxic hepatitis in 3 patients, progressive familial intrahepatic cholestasis (PFIC) in 3 patients, liver hemangioma in 3 patients, parenteral nutrition-associated liver disease in 3 patients and other in 5 patients. In one patient, the indication for liver transplantation was Alagille syndrome, in which kidneys may be affected in the form of renal dysplasia; however, this patient did not present renal dysfunction before LT. In seven patients, liver disease manifested as an acute liver failure (ALF), whilst chronic liver failure was present in 49 patients. One patient developed acute-on-chronic liver decompensation. Living-related donor liver transplantation (LDLT) was performed in 58 patients, and 3 patients underwent deceased donor liver transplantation (DDLT). Initial immunosuppressive therapy was based on tacrolimus (TAC) in 100% of patients and included: TAC + corticosteroids in 51% of patients and TAC + MMF in 36% and TAC + MMF + corticosteroids in 13%. Induction therapy (anti-IL2R ab.-basixilimab or daclizumab) was used in 15% of patients. One patient was converted to cyclosporine due to TAC intolerance. During follow-up, four patients underwent liver re-transplantation. Clinical and demographic data are summarized in Table 1.

### 3.2. Assessment of Renal Filtration Function

Since the results of cystatin C measurements were not fully available in the retrospective part of the study, eGFR was calculated based on the modified Schwartz “bedside” formula. Pre-transplant mean eGFR = 131 mL/min/1.73 m^2^ (±63 mL/min/1.73 m^2^) decreased to 109 mL/min/1.73 m^2^ (±56 mL/min/1.73 m^2^) at discharge after LT, then increased at 1 and 5 years after LT to 113 mL/min/1.73 m^2^ (±45 mL/min/1.73 m^2^) and 133 mL/min/1.73 m^2^ (±41 mL/min/1.73 m^2^), respectively. eGFR deteriorated afterwards to 124 mL/min/1.73 m^2^ (±36 mL/min/1.73 m^2^) at 10 years after LT and to 113 mL/min/1.73 m^2^ (±30 mL/min/1.73 m^2^) at >10 years of follow-up after LT. Comparison of all eGFR values showed statistical significance with *p* < 0.05 with the exception of values after LT and 1 year after LT with *p* = 0.4. The changes in renal function during follow-up are presented in Figure 1. To visualize the proper evolution of CKD stages in our cohort, we created a graph showing the percentage of patients belonging to each CKD stage during follow-up (Figure 2).

The number of patients with eGFR < 90 mL/min/1.73 m^2^ has been rising gradually from 10 years after LT onwards, and 46% of patients were in CKD stage 2 in the last follow-up after LT, according to eGFRcys values.

Creatinine-based estimations of eGFR tend to overestimate filtration rate; therefore, the consecutive values of eGFR calculated >10 years after LT in the planned analysis part of the study were based on a more precise cystatin C, urea and creatinine equation. The mean value of eGFR with the use of this formula was 90 mL/min/1.73 m^2^ (±18 mL/min/1.73 m^2^). Regardless of the measurement method, the values of eGFR < 60 mL/min/1.73 m^2^ (below a threshold of CKD stage 2) were present in two patients (3%) only at the end of follow-up. None of the patients developed CKD stage 5 (equivalent to end-stage kidney disease). Patients requiring renal replacement therapy before LT improved renal function during the perioperative period and did not develop CKD in long-term follow-up (presenting mean eGFRcys 95 mL/min/1.73 m^2^; ±20 mL/min/1.73 m^2^). Kaplan–Meier analysis using the threshold of eGFR < 60 mL/min/1.73 m^2^ as the event is presented in Figure 3. Two patients with eGFR < 60 mL/min/1.73 m^2^ 1 year after LT improved their eGFR levels up to 5 years after LT, and afterward, their eGFR has gradually deteriorated to values of 75 and 61 mL/min/1.73 m^2^. Patients reaching eGFR < 60 mL/min/1.73 m^2^ at final follow-up experienced gradual deterioration of their eGFR values from 5 years after LT onwards.

We have also compared the serum creatinine and urea concentrations and values of eGFR between patients with eGFR < 90 and >90 mL/min/1.73 m^2^ (by cystatin C-based formula) in long-term (>10 years) follow-up. No differences were found between the study groups in terms of eGFR values before LT, the etiology of liver failure, the nature of the progression of liver failure (acute vs. chronic), the number of patients on dialysis before LT and the number of patients with concomitant renal dysfunction before LT. However, the serum creatinine and consequently the consecutive eGFR values were regularly and significantly lower in patients from a group of lower final eGFR values (<90 over 10 years), at 1, 5 and 10 years post-transplant, respectively. The results are shown in Table 2.

### 3.3. Immunosuppression and Renal Function after LT

According to widely accepted standards, the doses of calcineurin inhibitors (CNI) should be gradually reduced in long-term follow-up. The mean value of trough (C_0_) TAC blood concentration during post-transplant follow-up was lowered from 9.6 ng/mL (±2.5 ng/mL) to 3.9 ng/mL (±1.8 ng/mL) at 10 years after LT. Trough TAC levels are shown in Figure 4.

Details of immunosuppression therapy at the last follow-up are shown in Table 3.

As the majority of patients received TAC as basic maintenance immunosuppressive, further analyses were aimed to evaluate associations between TAC C_0_ and eGFR.

Patients with TAC C_0_ > 10 ng/mL at hospital discharge presented lower eGFR values (82 vs. 105 mL/min; *p* < 0.006), higher creatinine (0.4 vs. 0.3 mg/dL; *p* < 0.002) and urea concentrations (40 vs. 33 mg/dL; *p* < 0.03), compared to patients with TAC_0_ < 10 ng/mL. Initially, the two groups did not differ in terms of the pre-transplant value of eGFR; however, presenting TAC C_0_ > 10 ng/mL at discharge was not associated with a further decrease of eGFR at long-term follow-up. After analyzing several variables of the study, a Cox proportional hazard model was created to evaluate the risk of lowering eGFR < 90 mL/min at >10 years of follow-up. The analysis revealed that patients older than 2 years at transplantation and those with tacrolimus levels higher than 4 ng/mL in distant follow-up are at greater risk of lowering eGFR < 90 mL/min/1.73. The results are shown in Table 4.

In a follow-up after >10 years, patients with blood TAC C_0_ > 4 ng/mL (median value for the group) had lower serum albumin (42 vs. 46 mg/dL; *p* < 0.04), lower total protein levels (70 vs. 74 mg/dL; *p* = 0.06) and higher urinary protein loss as measured by albuminuria (21 vs. 5 mg/dL; *p* < 0.02). Data from analysis >10 years after LT are shown in Table 5.

## 4. Discussion

The presented study is based on the experience of a pediatric transplant center and long-term observations of children after liver transplantation. We analyzed data and outcomes of 61 patients after liver transplantation with at least 10-year observation. Before liver transplantation, a total of nine patients (15%) had eGFRcr values below 59 mL/min or required renal replacement therapy. These observations are consistent with reports available in the literature [10] and emphasize the important role of liver function in renal homeostasis. It is important to analyze whether pre-transplant inferior renal function is an independent risk factor of further development of chronic kidney disease (CKD) over time. There was no such correlation in our patients. One of the reasons might be that none of the patients presented CKD before liver transplantation, and only four patients (6.5%) required continuous venovenous hemodialysis because of acute kidney injury (AKI), secondary to acute liver failure (ALF). None of these four patients had a diagnosis of renal disease before developing ALF. Two of them also required renal replacement therapy immediately after LT; however, renal function in patients requiring hemodialysis before LT was not significantly different from the other children (non-AKI patients) at 1 year, 5 years, 10 years and at final follow-up. There are several data on the significant impact of AKI on patient survival after LT and long-term renal function in survivors [11,12]. The results of the retrospective study of 102 children after LT indicated that patients requiring renal replacement therapy in the perioperative period presented inferior renal function at 10 years after LT, compared with non-AKI patients (eGFR mean 99 mL/min vs. 122 mL/min; *p* = 0.06) [13]. Our experience shows a distinct outcome in this term; however, due to the limited number of relevant cases, our observation may not be conclusive. In the post-LT period, a statistically significant, step-wise reduction in eGFR values from baseline was observed in the entire study group (median 91 vs. 130 mL/min/1.73 m^2^), and 48% of patients had eGFR < 90 mL/min (CKD stage 2 and below), including three patients (5%) with eGFR values between 45 and 59 mL/min (CKD stage 3). The deterioration of native renal function directly after LT is well described in the literature [4,6] and is mainly attributed to the acute nephrotoxic effect of calcineurin inhibitors (CNI) and the occurrence of AKI [4]. We did not find any correlation between other known risk factors of kidney damage in the postoperative period, such as the patient’s age at the time of transplantation, the patient’s gender or the etiology of liver failure in the study group.

There was neither a correlation between the duration of the anhepatic phase, ranging from 52 to 183 min (median 105 min), with the further, early or late deterioration of renal function, at hospital discharge, 1 year, 5 years, 10 years and at long-term follow-up. There are reports on the association between the duration of the anhepatic phase and the incidence of AKI [14] in liver transplant patients, but its impact on long-term renal function has not yet been evaluated. Our study did not show any effect of the duration of the anhepatic phase on the incidence of CKD, nor on the reduction of eGFR < 90 mL/min in the long-term follow-up.

The nephrotoxic effect of calcineurin inhibitors (CNI), including tacrolimus, has been widely reported in the literature both in the initial period after liver transplantation and in the long-term follow-up [15]. The adverse effects of CNI on renal function are related to afferent arterial vasoconstriction, resulting in decreased glomerular filtration capacity and (in a separate pathway) the direct damage of renal tubules. Both effects are dose/blood concentration-dependent and reversible in the acute phase [15]. Long-term exposure to CNI increases the risk of chronic kidney damage, finally leading to renal fibrosis.

The results of the study group confirm that the blood concentration-dependent effect of tacrolimus on eGFR is a case, as was reflected both in a positive correlation of serum urea and creatinine with blood TAC C_0_ and in lower eGFR values at hospital discharge in children with a drug concentration >10 ng/mL. The lack of effect of TAC C_0_ concentration at hospital discharge on long-term renal function suggests that early association was transient and could be managed with further relevant dosing procedures of tacrolimus. Immunosuppression protocol used in our center, based on consecutive tapering of TAC dose/concentration, is aimed at the range of TAC C_0_ between 6 and 8 ng/mL during 3 months after LT and to <6 ng/mL after 1 year. These nephroprotective strategies that minimize exposure to TAC after LT showed a beneficial effect on renal function without decreasing long-term survival. Apparently, such protocol is suitable in patients of low immunological risk. The other issue is the lowest effective range of TAC concentration. In one retrospective study evaluating the long-term survival of 235 adult liver recipients in regard to the mean values of tacrolimus concentration at 4 weeks after LT, there were no differences in survival or in the incidence of acute rejection episodes between groups with drug concentrations of 5–7, 7–10 and 10–15 ng/mL; however, patients with TAC C_0_ < 5 ng/mL presented inferior survival [16]. In the above-mentioned groups, the most common cause of death was tumor progression in 47% of patients, and graft loss was the cause of death in 6% of patients.

In general, in our entire group, eGFR improved 1 year after LT, although the change in median value did not reach statistical significance (91 vs. 107 mL/min; *p* = 0.4). However, there was a significant decrease in the number of patients with eGFR values <90 mL/min (17 vs. 29 patients; *p* < 0.03) over time. None of the examined children was diagnosed with CKD stage 4 or 5. At the same time, the median blood concentration of TAC during the study period decreased from 9.4 to 6.45 ng/mL (*p* < 0.000001). Considering the significant adverse effect of tacrolimus concentration on eGFR values in the postoperative period, it can be assumed that further improvement in renal function was associated with minimized exposure to the nephrotoxic effects of TAC, while the above-described target range of TAC C_0_ over time seems to be effective in terms of liver graft protection. After analyzing the eGFRcr values over the whole study period, the most noteworthy fact is a significant deterioration of renal function immediately after LT, a gradual improvement in eGFR up to 5 years after LT and then a systematic decline in their function >5 years after liver transplantation.

Gradual improvement of post-transplant eGFR seen in our patients was also described in other reports [17,18,19]. Some investigators emphasize that the deterioration of renal function during this period is particularly noticeable when cyclosporine A is used as the basis of immunosuppression [19]; however, this suggestion may be biased by dosing practice (and maintaining of specific target blood concentrations) at the individual center. In the study group, only one patient (1.5%) was taking cyclosporine (after conversion from TAC); therefore, no conclusion is possible in this term.

Five years after liver transplantation, we found improved renal filtration function in the entire study group. None of the patients developed stage 4–5 CKD, and only 10% had eGFR values <90 mL/min. At the same time, tacrolimus-based immunosuppression monotherapy was used in more than 70% of patients, with median TAC C_0_ concentrations of 3.4 ng/mL. These results are consistent with data available in the literature that evaluated children with a comparable immunosuppression protocol [4,20]. Arora-Gupta et al. report that in a group of 113 children after LT, despite an initial decrease in GFR that was associated with high doses of CNI, long-term therapy with lower doses of CNI was not associated with further deterioration of renal function 5 years after LT [20]. Again, it should be stressed that such low exposure to immunosuppression is suitable and safe only in stable and low-risk patients.

Naesens et al., in a comprehensive review of the literature on the nephrotoxicity of calcineurin inhibitors in organ transplant patients [21], noted that at >10 years post-liver transplantation, all renal structures in native kidneys could be irreversibly damaged due to chronic nephrotoxicity of CNI. Myers et al. were the first to show that cyclosporine not only induces reversible changes in renal vascular resistance but is associated with irreversible damage to renal architecture in native kidneys. In this study, interstitial fibrosis and tubular atrophy accompanied by focal glomerular sclerosis were described in native kidney biopsies of heart transplant recipients [1]. In our group, only two patients (3%) developed stage 3 CKD. We did not perform native kidney biopsies, as there was no clinical indication to do so.

Proteinuria may accompany a decreased renal filtration in CKD and occur in any of its stages. The test that has been incorporated into the Kidney Disease Improving Global Outcome guidelines (Albumin-to-creatinine ratio-ACR) uses the ratio of albumin to creatinine concentrations in a single urine sample. According to Gansevoort et al., the prognostic value of urinary albumin measurement in predicting albumin excretion in a 24 h urine collection is satisfactory and comparable to the albumin/creatinine ratio [22]. In the study group, an increased urinary albumin excretion was found in 25% of patients and ranged from 20 to 645 mg/L with a median value of 53.5 mg/L. However, there was no statistically significant association between urinary albumin concentrations and values of eGFRcr or eGFRcys. On the other hand, there was a significant correlation between albuminuria and tacrolimus concentration in the blood. Patients with TAC C_0_ > 4 ng/mL in the long-term follow-up had significantly higher proteinuria (21 vs. 5 mg/L) and lower serum albumin levels (43 vs. 45 mg/dL) compared with patients with tacrolimus concentrations <4 ng/mL. A statistically significant inverse correlation of tacrolimus concentration to serum albumin was also observed. This association is not clear, as the degree of proteinuria observed in our patients was too low to explain the lower albuminemia. Despite the significant difference, the direct values of albumin concentration are not clinically relevant (43–45 mg/dL); therefore, this observation may not be conclusive.

Currently, a number of authors consider albuminuria to be a GFR-independent indicator of renal filtration dysfunction [23,24]. In addition, there are reports describing an increase in albuminuria with increasing eGFR in the general population, which investigators explain by the phenomenon of nephron hyperfiltration [25]. Microalbuminuria can be used as a biomarker for the early screening of kidney damage in children after organ transplantation due to nephrotoxicity induced by calcineurin inhibitors and can predict further overt proteinuria and development of severe nephropathy [26]. Taking into account that in the studied group, a significant percentage of patients had elevated urinary albumin concentration, it seems reasonable to routinely determine ACR index during follow-up visits in children after LT. This may allow an earlier and better assignment of patients to the risk group of CKD complications (even despite eGFR values within normal limits): low, moderate, high and very high according to KDIGO [27] criteria. This may also allow an early introduction of nephroprotective treatment, including blockade of the renin-angiotensin system with specific drugs (ACEi).

Our study has several limitations. The first one is related to the fact that a part of the study is of a retrospective nature. The second limitation is a relatively small cohort size which, on the other hand, is difficult to overcome in the pediatric population with long-term follow-up analyzed in the single-center study. Another potentially confounding factor is related to the exclusion of the patients with fatal outcomes after liver transplantation and patients who presented early stage of AKI during transplantation procedure. On the other hand, a long-term follow-up was limited to a more homogenous population.

In conclusion, kidney function in children after liver transplantation tends to deteriorate steadily over 5 years after transplantation, which may lead to the development of chronic kidney disease in adulthood. The main factor that worsens renal function in children both in the early period after liver transplantation and in long-term follow-up is the intake of calcineurin inhibitors. The use of immunosuppression protocols with minimization of their dosage already in the first years after transplantation has an important nephroprotective effect. Microalbuminuria detected in a significant proportion of children at a distant time after liver transplantation appears to be an early, subclinical risk factor for worsening renal function. The introduction of urine albumin to creatinine ratio (ACR) testing during outpatient visits in selected periods after liver transplantation may enable the early identification of patients at risk of developing chronic kidney disease.

## Figures and Tables

**Figure 1 children-08-00633-f001:**
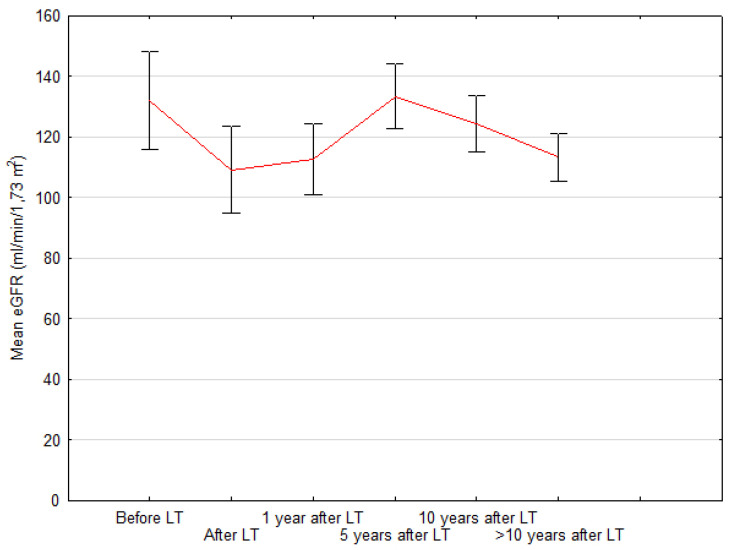
The evolution of eGFR. LT- liver transplantation.

**Figure 2 children-08-00633-f002:**
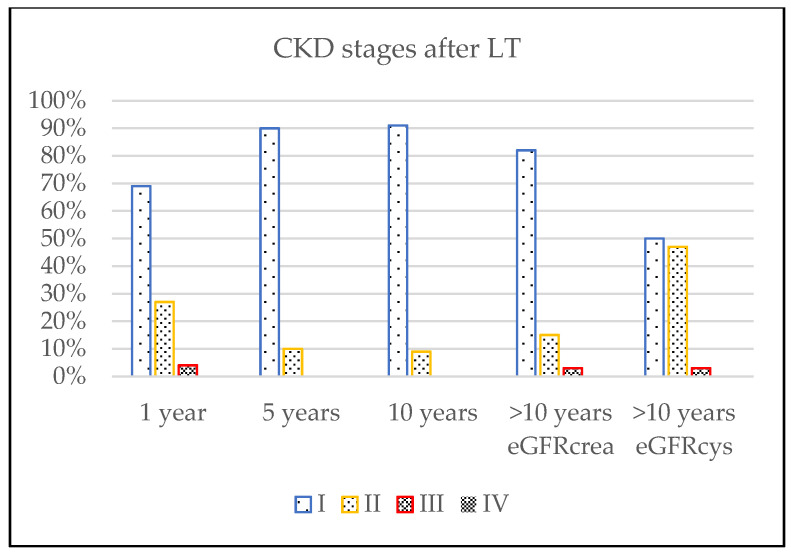
The evolution of Chronic Kidney Disease (CKD) stages.

**Figure 3 children-08-00633-f003:**
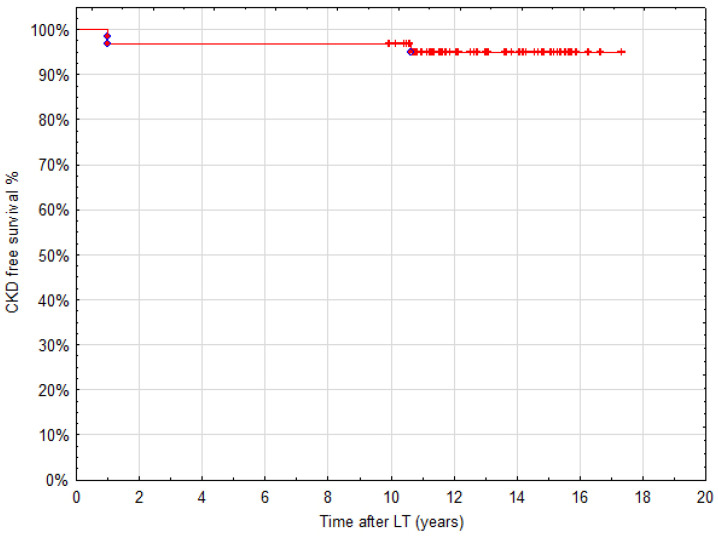
CKD-free survival.

**Figure 4 children-08-00633-f004:**
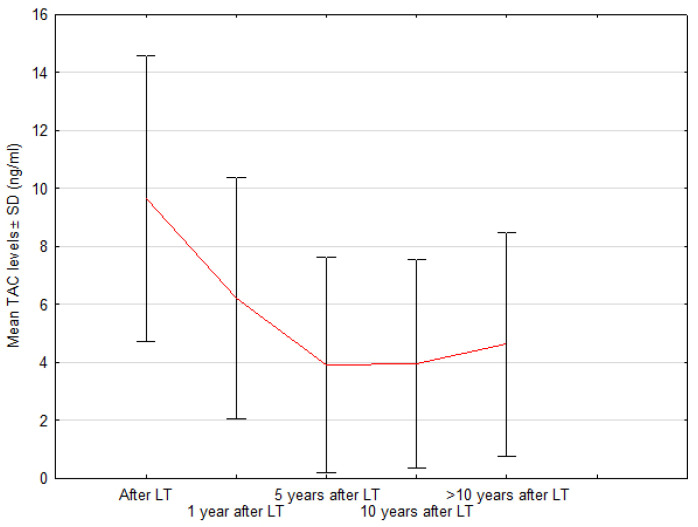
Mean tacrolimus (TAC C_0_) blood concentration during post-transplant follow-up.

**Table 1 children-08-00633-t001:** Baseline characteristics of liver recipients.

	*n* = 61 Mean ± SD	Median (IQR)
Age at transplant (years)	1.7 (±1.6)	1.2 (0.66–2.04)
Height (cm)	79 (±14.4)	75 (70–86)
Body mass (kg)	10.2 (±4.4)	9.8 (7–12)
Gender (male/female)	19/42	
Donor type (living/deceased)	58/3	
Liver failure (patients, %) ALF ChLF Acute-on-chronic liver failure Liver tumors	7 (11%) 49 (80%) 1 (2%) 4 (7%)	
PELD/MELD	17 (±11)	18 (11–20)
Creatinine (mg/dL)	0.3 (±0.2)	0.2 (0.2–04)
Urea (mg/dL)	23.7 (±12.5)	21 (15–29)
Creatinine-based estimated GFR, mL/min/1.73 m^2^	132 (±63)	130 (93–155)
Continuous veno-venous hemodiafiltration (patients, %)	4 (6.6%)	

ALF (acute liver failure), ChLF (chronic liver failure), PELD (pediatric end-stage liver disease), MELD (model for end-stage liver disease), SD (standard deviation), IQR (interquartile range).

**Table 2 children-08-00633-t002:** Comparison of patients with eGFR < 90 and > 90 mL/min/1.73 m^2^ > 10 years after LT.

	eGFR_cys_ > 90 mL/min/1.73 m^2^	eGFR_cys_ < 90 mL/min1.73 m^2^	*p* Value
Creatinine (mg/dL)	*n* = 30	*n* = 31	
1 year after LT	0.3	0.4	<0.02
5 years after LT	0.32	0.44	<0.0003
10 years after LT	0.45	0.59	<0.000001
Urea (mg/dL)			
1 year after LT	22	28	<0.007
5 years after LT	19	26	<0.02
10 years after LT	20	27	<0.002
eGFR_cr_ (mL/min/1.73 m^2^)			
Before LT	127	137	=0.36
At discharge after LT	98	91	=0.26
1 year after LT	115	90	<0.04
5 years after LT	141	106	<0.0007
10 years after LT	134	103	<0.000001
Age at LT (years)	1.00	1.33	=0.07
Acute liver failure	4 (13%)	3 (10%)	=0.65
Dialysis before LT	2 (7%)	2 (6%)	=0.97
Intensive care unit stay (days)	7	4	=0.09
Anhepatic phase duration (min)	105	105	=0.74

**Table 3 children-08-00633-t003:** Immunosuppression >10 years after LT.

	*n* = 61	%
Tacrolimus	47	77.5%
Sirolimus	5	8%
Cyclosporine A	3	5%
Mycophenolate mofetil (MMF)	2	3%
Corticosteroids	2	3%
Tacrolimus + sirolimus	1	1.5%

**Table 4 children-08-00633-t004:** Factors associated with lowering eGFR < 90 mL/min/1.73 m^2^ in Cox proportional hazard model.

Variables	Unadjusted HR (95% CI)	*p*-Value	Adjusted HR (95% CI)	*p*-Value
Age <2 y at LT	0,27 (0.12–0.6)	0.001	0.15 (0.05–0.41)	0.0003
eGFR before LT	1 (0.99–1.00)	0.7	-	-
eGFR 1y after LT	1 (0.99–1.01)	0.47	-	-
Tac_0_ concentration <4 ng/mL in distant follow-up	0.51 (0.21–1.22)	0.13	0.36 (0.14–0.92)	0.03

HR—hazard ratio, CI—confidence interval, Tac—tacrolimus.

**Table 5 children-08-00633-t005:** Results of analysis >10 years after LT.

	Range	Mean; ±SD	Median; (IQR)
Follow-up (years)	10–17.3	12.9; ±2	12.7; (11–14)
Creatinine (mg/dL)	0.33–1.3	0.63; ±0.2	0.6; (0.5–07)
Urea (mg/dL)	10–103	26; ±12	24; (21–30)
Cystatin C (mg/L)	0.53–2.14	0.92; ±0.27	0.87; (0.75–0.99)
Total protein (g/L)	13.8–88.5	70; ±11	72; (68–76)
Albumin (g/L)	32–49	43; ±4	42; (35.6–44.6)
Microalbuminuria (mg/L)	3–646	40; ±106	7.9; (4.1–23.2)
eGFRcr (mL/min/1.73 m^2^)	49–199	113; ±30	111; (94–131)
eGFRcys (mL/min/1.73 m^2^)	41–150	90; ±18	89; (80–101)

## Data Availability

Most relevant data are within the paper. Most of the output data were taken from the Polish National Transplant Registry at https://rejestrytx.gov.pl/tx/ (accesed on 1 May 2021). Since the data collected in the Registry is sensitive and thus, protected by law (the Act on Personal Data Protection and the Medical Records Act), access to the database is limited; it can be accessed only upon meeting registration criteria. The Registry is under the supervision of the Polish Transplant Coordinating Centre “Poltransplant”, a budget-funded unit of the Polish Ministry of Health. Since the Registry is only available to a limited number of healthcare professionals working in transplantation units across Poland, access to the database is impossible for people not involved directly in the transplantation and coordination process on the national level. All of the patients’ medical histories and other vital information used in the creation of the database are available directly at the Children’s Memorial Health Institute, Warsaw, Poland, after contact with the Director of Scientific Affairs, Piotr Socha at P.Socha@ipczd.pl.

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
