# Peer review of "Long-Term Follow-Up of Renal Function in Children after Liver Transplantation—A Single Center Retrospective Study"

_children, 2021, doi:10.3390/children8080633_

Round 1

Reviewer 1 Report

I thank the authors for their submission on long-term kidney function among pediatric liver transplant recipients. The language of the manuscript is clear, a fact for which the authors should be congratulated. That said, the study has no clear objective of hypothesis. It includes a small number of patients transplanted over the course of a long period of time (and some transplanted as long ago as 30 years). Over the past 30 years, many changes in not only donor and recipient management and transplant technique have occurred.

Given the small, heterogeneous population of patients that were analyzed, it is hard to know if the findings of the study that conflict with previous publications are truly valid (pre-transplant AKI, for example, not associated with inferior renal function in the long term in this study, though there were only 4 cases among 61 with pre-transplant AKI). On the other hand, the finding that higher CNI trough levels at 1 year were associated with worse kidney function in the long term is not surprising in the least.

Overall, this manuscript adds little to the medical/pediatric transplant literature and is unlikely to influence clinical practice.

Minor comments:

While study period belongs in Methods (should be "Patients and Methods" as opposed to "Materials and Methods"), the number of transplants performed during the period belongs in Results. Range of follow-up on individual recipients also belongs in Results.

Author Response

Dear reviewer:

I would like to thank you for taking the time to assess our article. We greatly appreciate the thorough and thoughtful comments provided on our submitted article. Your comments significantly improved our manuscript. It has taken some time to complete the final revision so thank you for your patience. We made sure that each one of the comments has been addressed carefully and the paper is revised accordingly.

Attached below are detailed responses to the reviewer’s comments. The letter is  shown in black and our responses in red. Please let us know if you still have any questions or concerns about the manuscript. We will be happy to address them, now in a timely manner.

Sincerely,

The authors

I thank the authors for their submission on long-term kidney function among pediatric liver transplant recipients. The language of the manuscript is clear, a fact for which the authors should be congratulated. That said, the study has no clear objective of hypothesis. It includes a small number of patients transplanted over the course of a long period of time (and some transplanted as long ago as 30 years). Over the past 30 years, many changes in not only donor and recipient management and transplant technique have occurred.

Our study consists of two parts – a prospective one and a retrospective one. The population of our study cohort was recruited from a group of patients with more than 10 years of follow-up after liver transplantation, remaining under care of our outpatient clinic and willing to  participate in our study after receiving an informed consent. As a pediatric transplant center, we are  limited  to treat the patients under 18 years of age and the earliest transplantation (included in the analysis) was performed in 2001. Prospective analysis was performed for data available between 2017 and 2019. We recognize that our methods were not explained properly and this was a reason of the misunderstanding. We have substantially modified the “patients and methods” section to resolve this issue.

Given the small, heterogeneous population of patients that were analyzed, it is hard to know if the findings of the study that conflict with previous publications are truly valid (pre-transplant AKI, for example, not associated with inferior renal function in the long term in this study, though there were only 4 cases among 61 with pre-transplant AKI). On the other hand, the finding that higher CNI trough levels at 1 year were associated with worse kidney function in the long term is not surprising in the least.

Our population consisted of 61 patients transplanted from 2001 to 2008, mainly young children (median age at LT was 1,2 years) who underwent liver transplantation from a living related donor in 95% of cases. In 80% of cases indications for transplantation was due to a chronic liver disease. 100% of our patients received tacrolimus as a main immunosuppressive agent at the time of LT. The only change related to immunosuppression protocol involved the use of corticosteroids which were initially used as a part of a standard triple immunosuppression regimen and after 2006 became limited to individual indications.  According the comment on the population size  -  it was  relatively small indeed, however this kind of a  limitation is typical and really difficult to overcome in pediatric transplant centers. Our report  is a single -center study, so the numbers (as per single center) could not be higher. Another fact is that when analyzing long term follow-up after LT  it should be stressed, that many  patients are transferred to adult transplant centers after reaching 18 years of age. Similar problems with a sample size have been faced by the other authors,  such as  Kivela et. al. [2011[1]] published in Transplantation, who reported a series of 57 patients with long term follow up of kidney function in children after LT.

 In regard to the  to risk factors of CKD: the use of calcineurin inhibitors (CNI) is a well-established one, however  the immunosuppression protocols are different between the transplant centers worldwide. Some  of them use mainly cyclosporine A, while the others  - tacrolimus. The optimal exposure to the CNI also remains a subject of discussion. Our immunosuppression protocol is based on the use of tacrolimus after LT with initial trough levels of 10-12 ng/ml and a significant stepwise reduction to 4-6 ng/ml 12 months after LT. This approach was uniform throughout the whole group and might result in the low incidence of CKD in our cohort,  in contrary to the   results  of the other published studies, reporting a higher incidence of CKD after LT (up to 30% in some series with a sample size ~100 patients, mainly  on cyclosporine immunosuppression[2]). Our approach was addressed in the “discussion” part of the manuscript. Moreover, the data on incidence of albuminuria as a marker of subclinical kidney injury after LT are limited, such as on its correlation with the use of CNI and measured cut-off points – and our study provides these information.

 In regard to the impact of acute kidney injury (AKI) : most of the studies report on incidence and short-term effects such as hospital mortality, need of vasoactive support and changes in laboratory findings, etc [3,4] or come from adult transplant centers [5-7]. On the other hand  the recent study analyzing the effect of AKI on the incidence of CKD in children after liver transplantation reports on a retrospective cohort with a high incidence of acute liver failure (20% in general group, 35% in AKI group) and small group of patients with biliary atresia and other chronic indications for LT (which obviously might influence both patient survival and long term kidney function)[8]. Apparently the selection of AKI criteria impacts the reported incidence. In our study,  AKI was only reported as a kidney failure requiring hemodialysis. Other (early) stages of AKI (as according to pRIFLE criteria) were not a subject of analysis.

Overall, this manuscript adds little to the medical/pediatric transplant literature and is unlikely to influence clinical practice.

Taking into consideration above provided explanations we hope that our manuscript might add some new information useful in  clinical practice, especially in terms of regular monitoring of albuminuria in children after liver transplantation.

 We recognize the pointed limitations of our study   –  a retrospective nature of the  part of the analyzed data, and a relatively small sample size. A limitations paragraph was added in the discussion part to address this issue.  

 To increase a value of the manuscript, we have modified the CKD stages presentation by adding additional figure in results section, we have added the median values and interquartile ranges for continuous variables and included Cox proportional hazard model in our analysis.

The manuscript content was also checked for the study design guidelines,  according to STROBE recommendations.

Minor comments:

While study period belongs in Methods (should be "Patients and Methods" as opposed to "Materials and Methods"), the number of transplants performed during the period belongs in Results. Range of follow-up on individual recipients also belongs in Results.

All of the above mentioned suggestions were addressed in the manuscript and inclusion criteria are properly disclosed in the “patients and methods” section.

  1. Kivela, J.M.; Raisanen-Sokolowski, A.; Pakarinen, M.P.; Makisalo, H.; Jalanko, H.; Holmberg, C.; Lauronen, J. Long-term renal function in children after liver transplantation. Transplantation 2011, 91, 115-120, doi:10.1097/tp.0b013e3181fa94b9.
  2. Campbell, K.M.; Yazigi, N.; Ryckman, F.C.; Alonso, M.; Tiao, G.; Balistreri, W.F.; Atherton, H.; Bucuvalas, J.C. High prevalence of renal dysfunction in long-term survivors after pediatric liver transplantation. J Pediatr-Us 2006, 148, 475-480, doi:10.1016/j.jpeds.2005.11.013.
  3. Ferah, O.; Akbulut, A.; Acik, M.E.; Gokkaya, Z.; Acar, U.; Yenidunya, O.; Yentur, E.; Tokat, Y. Acute Kidney Injury After Pediatric Liver Transplantation. Transplantation proceedings 2019, 51, 2486-2491, doi:10.1016/j.transproceed.2019.01.179.
  4. Hamada, M.; Matsukawa, S.; Shimizu, S.; Kai, S.; Mizota, T. Acute kidney injury after pediatric liver transplantation: incidence, risk factors, and association with outcome. J Anesth 2017, 31, 758-763, doi:10.1007/s00540-017-2395-2.
  5. Hilmi, I.A.; Damian, D.; Al-Khafaji, A.; Planinsic, R.; Boucek, C.; Sakai, T.; Chang, C.C.; Kellum, J.A. Acute kidney injury following orthotopic liver transplantation: incidence, risk factors, and effects on patient and graft outcomes. Br J Anaesth 2015, 114, 919-926, doi:10.1093/bja/aeu556.
  6. Junge, G.; Schewior, L.V.; Kohler, S.; Neuhaus, R.; Langrehr, J.M.; Tullius, S.; Kahl, A.; Frei, U.; Neuhaus, P. Acute renal failure after liver transplantation: incidence, etiology, therapy, and outcome. Transplantation proceedings 2006, 38, 723-724, doi:10.1016/j.transproceed.2006.01.074.
  7. Rueggeberg, A.; Boehm, S.; Napieralski, F.; Mueller, A.R.; Neuhaus, P.; Falke, K.J.; Gerlach, H. Development of a risk stratification model for predicting acute renal failure in orthotopic liver transplantation recipients. Anaesthesia 2008, 63, 1174-1180, doi:10.1111/j.1365-2044.2008.05604.x.
  8. Menon, S.; Pollack, A.H.; Sullivan, E.; Murphy, T.; Smith, J. Acute kidney injury and chronic kidney disease after non-kidney solid organ transplantation. Pediatric transplantation 2020, e13753, doi:10.1111/petr.13753.

Reviewer 2 Report

Comments for authors:

Kowalewski et al. present the result of a single-center retro-prospective study of renal function in children after liver transplantation. They found that CKD was diagnosed in about 3% of children, 12 years after LT, and blood tacrolimus level at 1 year after LT was an independent predictor of renal function decline after >10 years of follow-up.

  • The hypothesis of the study is not clearly explained. Overall the study is more descriptive. The authors should state the clear hypothesis towards the end of the introduction.
  • There is no clear description of how CKD was defined. This was defined as an eGFR <60 on the day of follow-up visit at 1, 5, and 10 years after LT. But there is no indication whether this was achieved with at least 2 or more values to confirm the sustained decline in eGFR from baseline. It should be mentioned as a limitation.
  • The outcome (CKD stages) are not clearly explained, for example how many transitioned from CKD stage 1 to stage 2 or further.
  • Based on the data presented, multivariate cox regression would be the better method to identify potential risk factors for developing the early stage CKD. The results should be displayed as a table in the manuscript.
  • The authors have not mentioned how they have censored the patients, also not clear if anyone died during the follow-up period, as death could be the competing risk factor.
  • The authors mentioned using the multivariate OR but it is not clear which adjusting covariates were included in the model.
  • In previous studies, eGFR <60 pre LT was an independent risk factor for the development of kidney outcome after LT. In this study, 15% of patients had eGFR <60 pre LT but none of them developed CKD. Not having enough samples could be one of the reasons to identify the signal or effect, and should be mentioned as a limitation.
  • It would be better to know the median (IQR) values in the baseline table for the continuous data.
  • Overall, manuscript content needs to be check for the study design guideline, such as STROBE.

Author Response

Dear reviewer:

I would like to thank you for taking the time to assess our article. We greatly appreciate the thorough and thoughtful comments provided on our submitted article. Your comments significantly improved our manuscript. It has taken some time to complete the final revision so thank you for your patience. We made sure that each one of the comments has been addressed carefully and the paper is revised accordingly.

Attached below are detailed responses to the reviewer’s comments. The original letter is  shown in black color and our responses in red.  

Sincerely,

The authors

Kowalewski et al. present the result of a single-center retro-prospective study of renal function in children after liver transplantation. They found that CKD was diagnosed in about 3% of children, 12 years after LT, and blood tacrolimus level at 1 year after LT was an independent predictor of renal function decline after >10 years of follow-up.

  • The hypothesis of the study is not clearly explained. Overall the study is more descriptive. The authors should state the clear hypothesis towards the end of the introduction.
  • The introduction part of our manuscript was modified to include clear hypothesis and research goals of our study

  • There is no clear description of how CKD was defined. This was defined as an eGFR <60 on the day of follow-up visit at 1, 5, and 10 years after LT. But there is no indication whether this was achieved with at least 2 or more values to confirm the sustained decline in eGFR from baseline. It should be mentioned as a limitation.
  • CKD was defined as eGFR below 60 ml/min per 1.73m2  according to Kidney Disease Improving Global Outcomes (KIDIGO)  and based on a single eGFR measurement which is now  clearly mentioned in the methods section.  

  • The outcome (CKD stages) are not clearly explained, for example how many transitioned from CKD stage 1 to stage 2 or further.
  • All of the CKD stages transitions are now reported as an additional figure in the results section.

  • Based on the data presented, multivariate cox regression would be the better method to identify potential risk factors for developing the early stage CKD. The results should be displayed as a table in the manuscript.
  • Cox proportional hazard model was added to our analysis and presented as an additional table.

  • The authors have not mentioned how they have censored the patients, also not clear if anyone died during the follow-up period, as death could be the competing risk factor.
  • Our study consists of two parts – a prospective one and a retrospective one. As the main objective was to analyze long-term kidney function, the cases of the deceased patients were not included in the analysis. The population of our study cohort was recruited from a living group of patients with more than 10 years of follow-up after liver transplantation, remaining under care of our outpatient clinic and willing to take care in our study after receiving an informed consent. Since, as a pediatric transplant center, we are only available to treat patients under 18 years of age the oldest transplantation was performed in 2001. Prospective analysis was performed between 2017 and 2019. We understand that our methods were not explained properly and thus the misunderstanding. We have substantially modified the “patients and methods” section to resolve this issue.

  • The authors mentioned using the multivariate OR but it is not clear which adjusting covariates were included in the model.
  • We have opted to use multivariate cox regression as you have previously suggested.

  • In previous studies, eGFR <60 pre LT was an independent risk factor for the development of kidney outcome after LT. In this study, 15% of patients had eGFR <60 pre LT but none of them developed CKD. Not having enough samples could be one of the reasons to identify the signal or effect, and should be mentioned as a limitation.
  • A limitations paragraph was added in the discussion part of our manuscript and we addressed the above-mentioned issue.

  • It would be better to know the median (IQR) values in the baseline table for the continuous data.
  • All the continuous data are now presented as a median value with IQR

  • Overall, manuscript content needs to be check for the study design guideline, such as STROBE.
  • We have checked our study with the STROBE guidelines checklist.

Round 2

Reviewer 1 Report

I thank the authors for the changes they have made to the manuscript. They provide a somewhat more clear objective, though there does not appear to be any hypothesis (i.e., specific concept they were hoping to address, other than just to report their center's experience).

The authors cannot call an analysis performed between 2017 and 2019 on patients transplanted between 2001 and 2008 "prospective". Similarly, in the text of the Discussion, they cannot say that "part" of the study was retrospective (whole study was retrospective).

I am somewhat unclear on the use of a Cox proportional hazards model to evaluate risk factors for developing eGFR <90, as eGFR may not necessarily progressively decline (i.e., can fluctuate). How did they evaluate time to this "event"? Did they wait until there were at least two (or more?) consecutive values <90? Did any patient's eGFR increase back above 90 at any point after falling below this value? Also, how did the authors create the model (what starting variables, what entry and/or elimination method, etc.)?

Author Response

Thank you again for taking the time to improve our manuscript. Below you can find responses to your comments.

I thank the authors for the changes they have made to the manuscript. They provide a somewhat more clear objective, though there does not appear to be any hypothesis (i.e., specific concept they were hoping to address, other than just to report their center's experience).

The authors cannot call an analysis performed between 2017 and 2019 on patients transplanted between 2001 and 2008 "prospective". Similarly, in the text of the Discussion, they cannot say that "part" of the study was retrospective (whole study was retrospective).

We agree that even though our analysis was part of a planned evaluation the study shouldn’t be called prospective. We have included the suggested changes in the manuscript.

I am somewhat unclear on the use of a Cox proportional hazards model to evaluate risk factors for developing eGFR <90, as eGFR may not necessarily progressively decline (i.e., can fluctuate). How did they evaluate time to this "event"? Did they wait until there were at least two (or more?) consecutive values <90? Did any patient's eGFR increase back above 90 at any point after falling below this value? Also, how did the authors create the model (what starting variables, what entry and/or elimination method, etc.)?

We have shown that the eGFR values fluctuate both in Figure 1. (as a median eGFR) and in Figure 2. (as a CKD stage). Since our goal was to evaluate long term kidney function after LT, the creation of Cox regression model was based on patients with eGFR<90 at >1 year after LT. Out of all the observed patients with eGFR<90 at 5, 10, >10 years after LT only 1 improved eGFR (i.e. had eGFR values<90 at 5 years after LT and >90 at long term follow-up). Time to event was defined as a presence of eGFR values <90 at either 5, 10 or >10 years after LT. Taking this fact under consideration we took liberty to perform a Cox analysis. For the evaluation of prognostic factors, we used the Cox regression model for proportional hazards, by computing hazard ratios (HR) and corresponding 95% confidence intervals (95% CI).  For the multivariate model we selected the variables that presented HR with a significance level ≤0,05 and those which, while not reaching the previous criteria, were considered relevant in the literature. We have modified the presentation of the Cox regression to include both adjusted and unadjusted HR values. This comment is also included in methods section of our manuscript.

Reviewer 2 Report

Thank you for providing a revised version of the manuscript based on the comments.

Some minor comments:

  • The CKD stages were defined based on a single eGFR value and assuming that was the time used to censor the events but how many had eGFR consistently lower than 60 in the follow-up visits? Also, add this as a limitation in the study.
  • Not sure if adjusted or unadjusted HR was used? Please describe it in the method properly. If adjusted was used then describe the adjusting covariates.
  • In table 1, the median (IQR) section has wrongly displayed the IQR.
  • There is no explanation for a new figure 2 in the paper.
  • The conclusion should be the last paragraph of the discussion part and the limitation paragraph should be the second last not the last.

Author Response

Thank you again for taking the time to improve our manuscript. Below you can find answers to your comments.

  • The CKD stages were defined based on a single eGFR value and assuming that was the time used to censor the events but how many had eGFR consistently lower than 60 in the follow-up visits? Also, add this as a limitation in the study.
  • Two patients with eGFR <60ml/min/1.73m2 1 year after LT improved their eGFR levels up to 5 years after LT and afterwards their eGFR has gradually deteriorated to values of 75 and 61 ml/min/1.73m2. Patients reaching eGFR <60ml/min/1.73m2 at final follow-up gradually deteriorated their eGFR values from 5 years after LT onwards. This explanation is now included above the Kaplan-Meier curve in the manuscript.
  • Not sure if adjusted or unadjusted HR was used? Please describe it in the method properly. If adjusted was used then describe the adjusting covariates.
  • We have added all the data regarding Cox model creation in the methods section. We have also modified the Cox regression table to include both adjusted and unadjusted HR.
  • In table 1, the median (IQR) section has wrongly displayed the IQR.
  • We have corrected this issue
  • There is no explanation for a new figure 2 in the paper.
  • Figure 2 is now explained.
  • The conclusion should be the last paragraph of the discussion part and the limitation paragraph should be the second last not the last.
  • We have corrected the order of mentioned paragraphs in the manuscript